# How Successful Are Veterinary Weight Management Plans for Canine Patients Experiencing Poor Welfare Due to Being Overweight and Obese?

**DOI:** 10.3390/ani14050740

**Published:** 2024-02-27

**Authors:** Kim K. Haddad

**Affiliations:** VCA San Carlos Animal Hospital, 718 El Camino Real, San Carlos, CA 94070, USA; kkhaddad@aol.com

**Keywords:** canine, dog, overweight, obesity, weight loss, animal welfare, diet, nutrition, veterinary

## Abstract

**Simple Summary:**

Poor welfare attributable to overweight and obesity is considered one of the most significant welfare issues affecting companion animals. Veterinarians play an important role in providing nutritional counseling and weight management advice, but successful weight loss is difficult to achieve and maintain. This research examines the level of veterinary engagement and the success of veterinary management plans for overweight and obese canines over a five-year period. The electronic medical records of overweight or obese canines from four San Francisco Bay Area small animal hospitals were statistically analyzed for level of veterinary engagement, weight loss success, prescription weight loss diet use, and comorbidities. The results suggest that veterinary engagement is variable and, even at the highest level, is not adequate to successfully combat overweight or obesity or result in improved welfare. Overweight and obesity is one of the most important and preventable animal welfare issues we face, yet successful treatment and management remain elusive.

**Abstract:**

Overweight and obesity is one of the most significant health and welfare issues affecting companion animals and are linked to several serious medical conditions, reduced welfare, and shortened lifespan. The number of overweight and obese pets increases every year. Overweight and obesity are associated with multiple chronic diseases. Underlying causes include human-related and animal-related factors. Veterinarians encounter overweight and obese canine patients in daily practice and they play an important role in weight management. This research examines the level of veterinary engagement and the success of veterinary management plans for overweight and obese canines over a five-year period. Electronic medical records (EMRs) were collected for 500 canine patients assessed as either overweight or obese and statistically analyzed for level of veterinary engagement (VE), weight loss success, prescription weight loss diet (RX) use, and comorbidities. The average age at the beginning of the study was 61.5 months. A starting Body Condition Score (BCS) of 6 or 7 was most common (87.2%). Twelve different small and large dog breeds were most highly represented (61.1%). Average weight loss rates were low and more dogs gained rather than lost weight (56.5% vs. 43.5%). While VE is important, this study suggests current VE levels are not adequate to successfully combat overweight and obesity or result in improved animal welfare.

## 1. Introduction

Overweight and obesity is one of the most common chronic diseases resulting in reduced welfare, serious illness, and shortened life expectancy in companion animals [1]. Overweight and obesity, distinguished by the amount of excess weight, are defined as excess adipose tissue and occur when an imbalance between energy intake and energy consumption results in a positive energy balance. This imbalance has a substantial negative impact on the welfare of the animal [2]. While there is no universally agreed-upon definition, many believe obesity is a complex, preventable, and treatable clinical state and should be considered a disease [2,3,4,5]. Most agree that overweight is defined as 10–20% above ideal weight and like humans, pets are considered obese when they are 25–30% or more above their ideal weight [6,7]. Using the recommended 1–9 Body Condition Score (BCS) for companion animals, 1 is emaciated, 9 is morbidly obese, and a score of 5 is ideal [7,8,9,10]. The Global Pet Obesity Position Statement calls for a uniform definition of obesity and for obesity to be formally recognized as a disease. They define obesity as 30% or more above ideal weight, corresponding to a BCS of 8 or greater [11].

BCS scoring relies on the observer’s visual and physical assessment of the dog by feeling the ribs, hips, spine, and waist. BCS, though a widely used and valuable tool, is a subjective measure. To determine whether a dog is overweight or obese, a BCS score is recorded in combination with the current weight. Serial measures are necessary to monitor the degree of weight gain or loss and to adjust the weight management plan. Unfortunately, documentation of weight and BCS by veterinarians is inconsistent. One U.K. study looked at the records of 148 dogs over 12 months and found that 29% of medical records mentioned weight and only one BCS measurement was documented [12].

Dogs are popular pets and recent surveys estimate that there are 80 million in the U.S. [13], 12 million in the U.K. [14], 97.5 million in the E.U., and more than 470 million globally [15]. Rates of canine overweight and obesity are 56% in the United States [16] and 51% in the United Kingdom [14] and range from 22 to 44% globally [7]. Overweight and obesity are estimated to affect more than 200 million dogs globally, representing one of the most significant welfare issues of our time [1,7,11,12,13,14,16]. In 2020, the Pet Food Manufacturers Association study determined that 100% of 227 veterinarians surveyed were concerned about patient obesity, and 74% believed that the problem was growing [13]. Banfield, the largest veterinary practice in the U.S., reported a 108% increase in overweight and obese dogs in the past 10 years [17].

Overweight and obesity negatively impact the quality of life and longevity in dogs [7,18,19]. Overweight and obesity in dogs are associated with multiple health problems including cardiopulmonary and respiratory [20], orthopedic [21,22,23], dermatologic [24], endocrine [25], urinary [26], and neoplastic diseases [1,20,27], and behavioral problems [28]. As with humans, the most common diseases associated with overweight and obesity in dogs include orthopedic issues, such as osteoarthritis (OA) and degenerative joint disease (DJD), cardiovascular and respiratory diseases, diabetes mellitus, urinary incontinence and renal disease, dermatologic disease, neoplasia, and behavioral problems [2,18,21,29,30,31,32,33,34]. Studies show overweight and obese dogs are less active, less social, and more likely to exhibit undesirable behaviors and have reduced quality of life [19,28,29,35]. In addition to the comorbidities listed above, a U.K. study by German et al. determined that overweight dogs are more likely to display behavioral problems such as food guarding, stealing, and coprophagia (eating feces) [28]. Pet behavior problems are a significant welfare concern, and according to the American College of Veterinary Behaviorists, undesirable behavior is one of the top reasons animals are surrendered to shelters. Overweight and obesity result in poor welfare due in part to an increased risk of several other serious chronic diseases, reduced quality of life, and shortened lifespan by almost two years [18,21,31,32,33].

Causes of overweight and obesity in dogs have been broadly categorized as either animal-related or human-related [1,36]. Animal-related factors include predisposition, sex and reproductive status (intact/neutered), age, growth rate, and breed [1,37]. Genetic and behavioral factors may explain why some breeds are more prone to obesity [36]. The breeds at highest risk for obesity in the U.S. are Labrador retrievers, Golden retrievers, Cocker spaniels, Dachshunds, Dalmatians, Rottweilers, and Shetland sheepdogs [1], while in the U.K., Pug, Beagle, Golden retriever, and English springer spaniel breeds showed an increased risk of becoming overweight or obese [12]. The increased risk for these breeds may be due to lower energy requirements [1]. While data are limited on why some breeds are more prone to obesity than others, Raffan et al. determined that a deletion in the canine pro-opiomelanocortin (*POMC*) gene in Labrador retrievers is associated with weight and appetite in obesity-prone Labradors [38]. More research is needed to explore why some breeds are at higher risk for becoming overweight or obese.

Owner-related factors may be even more important than animal-related factors and owner behavior is considered a key factor influencing a dog’s weight [39]. Owners control the quality and quantity of food provided and the extent to which their pet exercises. Studies report owner feeding habits such as using food as a reward or to show affection, as well as lack of regular exercise, are some of the main contributors to overweight and obesity [1,23,40]. Feeding table scraps and over-treating are other challenges to successful weight management. Additionally, owners often fail to recognize when their dog is overweight [36,41,42,43]. According to the 2021 APOP Survey of pet owners, there remains a disconnect between owner perception of their pet’s weight and veterinary assessments [44].

There are many similarities between overweight and obesity in people and pets, with owner lifestyle playing an important role [10,20,34,45,46]. Owner risk factors that promote obesity in humans, such as weight, lifestyle, eating habits, and low levels of physical activity, mirror pet risk factors [47]. Using a One Health approach, a recent study of obese owners and dogs concluded that being an overweight dog owner was the most important factor in the occurrence of obesity in dogs [46]. Another human factor to consider is the impact of COVID-19. During the study period, COVID-19 lockdown restrictions resulted in changes in human behavior, and dogs were walked less often and for less time daily [48,49]. This may have also resulted in increased food intake by both humans and their dogs. A 2023 global study of factors associated with human weight gain during the pandemic found an increase in negative lifestyle habits such as an increased consumption of unhealthy foods and a decreased level of physical activity [50].

The role of the veterinarian is also very important when considering possible causes of overweight and obesity. According to a survey by the Veterinary Information Network (VIN), most veterinary appointments (53%) are 15 to 20 min and pets often present with multiple problems; discussing weight may be less of a priority during a routine wellness examination. [51]. One report argues veterinarians are not meeting their ethical and professional obligation to adequately address weight issues [47]. From a veterinary management perspective, time constraints, veterinarian lack of prioritization, and poor client compliance add to the challenges to preventing and treating this disease [10,52]. Diet history is an important part of medical history, but accurate information is rarely available, making nutritional analysis difficult. Equally important is poor client compliance once a weight management plan is presented. Client non-compliance is one of the most significant challenges to veterinary weight management program success [53]. Veterinarians may not prioritize weight management due to lack of time, lack of accurate diet information, and frustration from poor client compliance.

Treatment of overweight and obesity typically consists of feeding a calorie-restricted diet and encouraging increased physical activity. RX diets have been shown to result in clinically significant weight loss [54,55,56]. Increasing activity can increase energy expenditure to increase weight loss, improve cardiovascular function, and help maintain lean muscle mass during weight loss [57].

Several professional organizations have studied and reported on this disease and produced informative guidelines and tools for veterinary teams (AAHA, AVMA, Banfield, BSAVA, WSAVA, One Health Initiative). Links to this information can be found in Appendix A. But many of the suggested protocols and strategies are too time-consuming for the general practitioner to consistently implement as part of the wellness examination. For example, 2014 AAHA Weight management guidelines for dogs and cats recommend that veterinary weight loss programs include obtaining a detailed diet history, performing several calculations, making weight-related assessments, giving caloric intake recommendations, and developing feeding and exercise programs as part of the physical examination [10]. The AAHA recommendations are summarized in Table 1. Each step is important to the development of a safe weight management plan and requires a high level of veterinary engagement and a significant amount of time to complete.

The AAHA 2021 nutrition and weight management guidelines recommend that a nutritional assessment screening evaluation be performed on “every pet, every visit” [52]. The continued increase in disease prevalence suggests that these guidelines and tools, while useful, are not sufficient solutions to combat this complex and challenging medical and welfare problem [45,58,59,60,61]. Compared to humans, there are limited products and services available for canine weight loss and most support and guidance come from veterinarians. Obesity and overweight are a disease of humans and their dogs and any treatment and preventive measures must take this relationship into consideration. According to Day, “ The human-animal bond can serve as a mechanism to maintain motivation and adherence to physical activity and weight control strategies. Dogs provide social support for physical activity and [human] weight loss” [24] (p. 294).

Studies show that weight loss improves quality of life and lifelong caloric restriction can reduce disease incidence and improve longevity [18,21,22,29,30]. The benefits of weight loss are well established with improvements in various disease states, improved cardiovascular and respiratory function, reduction in pain from osteoarthritis, and reduction in endocrinopathies and chronic inflammation [22,23,25,59,60]. German et al. reported significant improvements in pain score, quality of life score, and vitality in dogs that had successfully lost weight and found that dogs that did not lose weight had increased emotional disturbances [18]. A review of the literature strongly indicates successful weight loss in dogs, while challenging, has a positive impact on health, welfare, and longevity. Successfully treating and preventing this disease requires addressing both human- and animal-related causes.

The aim of this study was to determine the extent and effectiveness of current veterinary weight management guidance by evaluating the level of VE. This study tests the hypothesis that increased VE results in more successful weight management. Additionally, the use of RX diets and weight-related comorbidities were analyzed by reviewing EMRs of canine patients diagnosed as overweight or obese between 2017 and 2022 from four VCA small animal veterinary hospitals in the San Francisco Bay Area, California. While several studies have looked at animal-related factors such as age, sex and reproductive status, and breed, as well as RX diet use and comorbidities, to this researcher’s knowledge, there are no published studies specifically evaluating VE.

## 2. Materials and Methods

### 2.1. Data Collection

Using VCA’s OSCAR Database search tool, a search of patient records was performed using the following criteria:Canine patients with either overweight or obesity listed as a concern or assessment;At least 12 months of age at the time of diagnosis;Male or female, altered or intact;Diagnosed in the period of 2017 to 2022;Medical records reporting a minimum of two weight and corresponding BCS measures.

The age criterion was set to include dogs above one year of age to eliminate the rapid increase in weight as expected in young growing dogs. While studies have shown the importance of weight management during the growth period, dogs less than 12 months of age were not included in this study. To prevent skewing weight and BCS results, dogs diagnosed with chronic, severe disease resulting in death or euthanasia shortly after their initial recorded diagnosis of overweight or obesity were excluded since many chronically ill patients suffer from inappetence or anorexia with rapid weight loss.

EMRs from four San Francisco Bay Area Hospitals were searched, resulting in 3400 canine patients. Using the Microsoft Excel^®^ randomization function, 500 unique patient ID numbers were selected for inclusion. No client information was included in the search. The decision to limit the study to 500 patient ID numbers was made based on the estimated time required to review the medical records and analyze the data within the time constraints of this research project. Seventeen patients with only one record and seven with chronic diseases were excluded. A total of 476 patients were included in this study. From these patients, a total of 2.437 EMRs were reviewed, in the period from 1 January to 24 August 2022. These medical records represented an average of 5.12 records per patient, where some patients had a minimum of two and a maximum of nine weight and BCS measurements. Each medical record was reviewed for weight, BCS, VE, RX diet use, and comorbidities.

### 2.2. Body Condition Scoring

The BCS for each dog was assessed using an established methodology which uses a 9-point scale to assess the body fat mass. Dogs included in the study had an initial BCS of 6 or higher. Patient BCS scores and associated weight for each veterinary exam were entered into an Excel spreadsheet up to a total of nine measurements. The BCS scores reported as decimals were rounded up to the next whole number. The EMRs reviewed utilize the standard SOAP format (Subjective, Objective, Assessment and Plan). BCS is listed along with other vital signs in the Objective section of the EMRs. Some veterinarians did not enter the BCS score in the designated field but rather in the physical exam findings. Weights without a corresponding BCS were not included.

### 2.3. Assessment of Veterinary Engagement (VE)

VE was determined from a review of the Plan portion of the EMR. The VE score was determined by the recommendations in the 2014 AAHA guidelines for weight loss with VE3 considered the level of engagement most consistent with these guidelines, as shown in Table 1 [10]. VE was categorized on a scale of 0–3, with 0 being no documented client communication regarding weight management and 3 being the most engaged as further described in Table 2. VE was scored based on the highest level of engagement as documented in all the medical records reviewed for each patient. There may have been several visits where the VE was scored 0–2, but if at any visit the VE was at the highest level (VE3), then that patient record would be recorded as VE3. The researcher acknowledges this is a limitation of this study and assessing VE at each visit would provide additional valuable information. Another limitation is that VE assessment is subjective and based on information provided in the EMR, which may not consistently or accurately reflect the actual level of VE.

### 2.4. Prescription Diet Use

Use of veterinary RX diets was determined by a review of the plan as well as a review of the prescriptions documented in the EMR. The use of other types of prescription diets such as for dermatologic and gastroenteric sensitivities or other medical conditions were not included in this study as they are not necessarily designed for weight loss. RX diet use was entered as either Yes or No and the date of the initial prescription was provided. There was no review of invoices to determine if RX diets were purchased from the veterinary hospital or elsewhere without preparing an actual RX. Additionally, in some cases, the duration of RX diet use was not documented. Unless a specific prescription was provided, the actual diet prescribed was not recorded. Client-described over-the-counter diets fed such as “less active”, “weight control”, “optimal weight”, or “healthy weight” were not included. These diets were eliminated since these claims are not recognized by the Association of American Feed Control Officials (AAFCO), which establishes the nutritional standards for complete and balanced pet foods, and they have a wide variety of caloric densities, not necessarily consistent with reduced-calorie food [54,55].

### 2.5. Comorbidity Assessment

Comorbidities were documented if they existed prior to or at any time during the study period. Comorbidities were grouped into the top disease categories commonly associated with overweight and obesity: orthopedic, endocrine, dermatologic, renal and urinary, cardiovascular and respiratory diseases, and neoplasia. VCA EMRs allow veterinarians to enter any description as an assessment; therefore, diagnoses do not have specific codes or require precise language. Different assessments were often made for the same condition. For example, chronic otitis externa and recurring otitis externa were both considered dermatologic diseases. A summary of the most frequently reported diseases and their corresponding categories is shown in Table 3.

### 2.6. Data Analysis

Data were collected from OSCAR and transferred into a spreadsheet using Microsoft Excel^®^ for Mac Version 16.63. Data were cleaned up and errors were corrected prior to statistical analysis. Excel was used to determine changes in BCS and weight, the frequency of RX use, and comorbidity occurrence and to assess the relationship between VE, BCS, and weight change. For all statistical analyses, the level of statistical significance was set at *p* < 0.05. The regression goodness of fit was determined with the Pearson r^2^ correlation coefficient and the statistical significance was determined via the F test.

Weight loss based on age, sex, and breed was analyzed using Excel functions to calculate mean, maximum, minimum, and median values. To evaluate the impact of veterinary engagement on weight loss, VE was categorized into 4 ordinal ranks (0–3). Each category of veterinary engagement (Table 2) corresponds to a different qualitative level of VE. The relationship between VE and BCS was analyzed by performing a linear regression of VE levels with respect to the initial BCS reported. The relationship between VE and weight change was analyzed by performing a linear regression of VE levels with respect to the observed weight change. The linear regressions performed in this study use the Ordinary Least Squares (OLS) method because the observations are independent and the residuals are normally distributed, homoscedastic, and linearly independent.

The effectiveness of RX diet use was assessed by calculating mean, maximum, minimum, and median values of weight change in dogs fed an RX diet compared to that of dogs not fed an RX diet. Comorbidity data were also analyzed. Because of the infrequency of incidence of some diseases in this sample size, for example, renal and urinary disease, the entire data set for these categories was analyzed rather than for each disease. Finally, BCS and sex were analyzed in relation to the incidence of comorbidities using linear regression to determine the relationship.

Supporting data in the form of an Excel spreadsheet are available by request.

## 3. Results

Medical records were reviewed for 500 canine patients with a BCS of 6 or higher and a diagnosis of either overweight or obese from 1 January 2017 to 24 August 2022. Seventeen were eliminated due to insufficient weight and BCS recording and seven were eliminated due to death within 6 months of the initial diagnosis. A total of 476 patients were included in this study. A range of ages, sexes, breeds, and BCSs were included in this study (Table 4 and Figure 1).

### 3.1. Veterinary Engagement

The results of the review of 2437 EMRs for VE are detailed in Table 5 The overall level of VE was determined to be highest at VE1 (48.7%; *n* = 232) and VE2 (31.3%; *n* = 149), with only 68 veterinarians engaged at the highest level VE3 (14.3%; *n* = 68), and 5.7% or 27 veterinarians did not document any client engagement in their EMRs (VE0).

For all levels of veterinary engagement, the mean weight change was greater than zero. The range of percent weight change was wide at all levels of VE. VE0, representing the lowest level of engagement, had the highest percentage of patients who lost weight (55.6%; *n* = 15), but 40.7% (*n* = 11) gained and 3.7% (*n* = 1) neither lost nor gained weight, with a mean change of +0.3%. VE1 had more pets who gained weight (53.9%; *n* = 125) than lost weight (44.4%; *n* = 103), and 1.7% (*n* = 4) neither lost nor gained weight, with a mean change of +2.5%. VE2 also had more pets who gained weight (55.7%; *n* = 83) than lost weight (43.6%%; *n* = 65), and 0.7% (*n* = 1) neither lost nor gained weight, with a mean change of +4.1%. Finally, at VE3, 60.3% (*n* = 41) of pets gained weight, 35.3% (*n* = 24) lost weight, and 4.4% (*n* = 3) neither lost nor gained weight, with a mean change of +4.9%.

VE and change in weight were analyzed to determine if increased VE correlated with more weight loss as a percentage of initial weight. Figure 2 below shows this relationship was very weak. Weight loss or gain as a percentage of initial weight at the time of diagnosis by VE is shown where the boxes represent the second and third quartiles. A review of the data revealed them to be normally distributed. Data points that fell outside of the Inter Quartile Range (IQR) by more than 1.5×IQR were examined as potential outliers. No data points were rejected as outliers.

To determine the statistical significance of the relationship between the initially diagnosed BCS and the corresponding VE, the null hypothesis—that there is no relationship between BCS and the level of veterinary engagement—was tested (Figure 3 and Figure 4). The observed value of F equals the critical value of F at the 96% probability level, which means that the null hypothesis is false. The observed value of F = 4.3 is greater than F_1,475_(0.05) = 3.9. The data suggest that the differences observed in levels of VE for overweight or obese dogs between BCS scores are statistically significant. The level of VE increases linearly with respect to the initial observed BCS, which was expected, but the degree of increase is small.

To determine the adequacy of VE, dogs were sorted into groups according to their respective maximum observed BCS. The total percentage weight change over the course of the study was then plotted against the level of VE (Figure 5). For each peak BCS, the correlation between the level of VE and percentage weight change was not found to be statistically significant. For dogs who reached a peak BCS of 6, this relationship was found to have an F value of (0.67), which equals the critical value of F at the 58% probability level; for BCS 7, a 2.80 F value was observed, equivalent to the critical F value at the 90% probability level; for BCS 8, a 0.01 F value was observed, equivalent to the critical F value at the 7% probability level; and for BCS 9, a 0.18 F value was observed, equivalent to the critical F value at the 33% probability level. The null hypothesis was determined to be true; therefore, for each BCS, an increase in VE did not yield statistically significant differences in weight change.

### 3.2. Prescription Weight Loss Diets

The number of dogs prescribed an RX diet during the study period was small (*n* = 40). Based on regression analysis, we determined that the relationship between the use of RX diets and weight loss was not statistically significant. RX diets were prescribed to 8.4% (*n* = 40) of dogs. The mean weight change of both groups of dogs, those on an RX diet and those not on an RX diet, was positive. The mean weight change of dogs on an RX diet (+0.9%) was less than the mean weight change of dogs not on an RX diet (+3.4%). A greater percentage of dogs on an RX diet lost weight (57.5%; *n* = 23) than dogs not on an RX diet (42.2%; *n* = 184). Although the number of dogs fed an RX diet was low, a greater percentage of dogs achieved an ideal BCS when fed an RX diet (15%; *n* = 6) compared to dogs not on an RX diet (10.1%; *n* = 44).

Although these results suggest RX diet use may result in greater weight loss, the correlation between RX diet use and percentage weight change was not found to be statistically significant. Testing of the null hypothesis, that there is no relationship between prescription diet use and percentage weight change, yielded an F value of (2.29), which equals the critical value of F at the 13% probability level. The null hypothesis was determined to be true, therefore prescription diet use did not result in statistically significant weight change.

### 3.3. Comorbidities

Comorbidities associated with a diagnosis of overweight or obesity were reported in 49.4% (*n* = 235) of dogs and 13.5% (*n* = 64) had more than one disease. More female (50.8%; *n* = 125) than male (47.8%; *n*= 110) dogs developed a comorbidity and females were more likely to have multiple comorbidities (15.9%; *n* = 39) than male dogs (10.9%; *n* = 25). Observed differences in comorbidity occurrence among male versus female dogs were not determined to be statistically significant. Orthopedic disease was the highest reported weight-related disease at 29.6% (*n* = 141), followed by dermatologic disease at 13.2% (*n* = 63), cardiovascular and respiratory diseases at 11.6% (*n* = 55), endocrine diseases at 4.4% (*n* = 21), neoplasia at 3.4% (*n* = 16), and renal and urinary diseases at 2.9% (*n* = 14). This distribution is displayed in Figure 6.

Comorbidity data were analyzed to test the hypothesis that sex is linked to whether overweight or obese dogs are more likely to develop any or multiple comorbidities. For any comorbidities, the observed value of F [0.42] equals the critical value of F at the 48% probability level, which means that the null hypothesis is false. The observed value of F [0.42] is less than F_1,475_(0.05) ≅ 3.86. The data suggest that the differences observed in overweight or obese dog comorbidity development between sexes are not significant. For more than one comorbidity, the observed value of F [2.54] equals the critical value of F at the 89% probability level, which means that the null hypothesis is false. The observed value of F [2.54] is less than F_1,475_(0.05) ≅ 3.86. The data suggest that the difference in multiple comorbidity development between sexes is also not significant (Figure 7).

## 4. Discussion

Overweight and obesity is one of the most common chronic diseases resulting in serious illness, reduced quality of life, and shortened life expectancy in companion animals [7]. More than half of all dogs in the U.S. and U.K. and anywhere between 22 and 44% of dogs globally are overweight or obese; based on recent dog ownership statistics, this suggests that more than 200 million dogs suffer from this disease [7,11,13,14]. Considering the number of animals affected, overweight and obesity may be considered the most significant and preventable welfare issues for companion dogs. Several studies have been conducted to determine the causes and health implications of the disease and benefits of weight loss, but to date, no studies have specifically analyzed the level of VE in relation to canine weight management. The results of this study show veterinarians do not consistently document a diagnosis of overweight or obesity; there is significant variability in BCS assessment; RX diets are infrequently recommended; and, regardless of the level of engagement, veterinarians have limited success in helping patients achieve their ideal body weight. This study also confirms the results of other studies showing a high incidence of comorbidities associated with overweight and obesity, with orthopedic diseases being the most reported. These results suggest that while overweight and obesity is a serious disease with multiple negative health consequences, veterinarians struggle to communicate and educate clients on the importance of weight management.

### 4.1. General Findings

Breed representation was an even mix of small-, medium-, and large-breed dogs. Labrador retrievers, German shepherds, and Golden retrievers topped the list of large breeds, mixed breeds and beagles were the most common medium breeds, and Chihuahuas and small mixed breeds were the most commonly overweight or obese small breeds. This is consistent with other studies of breeds at higher risk of being overweight or obese [1,12]. The data were not analyzed by specific breed size, but this might be of interest for further study. A broad range of ages was represented. Starting age ranged from 12 to 216 months with an average starting age of 62 months and average ending age of 98 months. A similar number of neutered female and neutered male dogs were represented, i.e., 49.6% and 46.4%, respectively, with a small percentage of dogs sexually intact (female 2.1%, male 1.9%). Spay and neuter rates in this geographic region are quite high and without weight data for a larger population of intact dogs, it is difficult to draw conclusions about the relationship between reproductive status and the diagnosis of overweight and obesity. But these results were consistent with other studies showing middle age and neutered status as animal factors commonly associated with overweight and obesity [1,12].

Overall, 43.5% of dogs lost weight from the initial diagnosis of overweight or obesity to the last recorded weight. Most dogs in this study failed to lose weight regardless of their initial BCS, weight, or level of VE. The average weight change for all dogs was +3.2%, the most weight lost was −28.7%, the most weight gained was +67.2%, and the median weight change was +2.0%.

One limit of this retrospective study is the difference in time intervals between weight and BCS assessments and data tracking weight variation during the study timeframe. Only initial and end weights were used to determine the overall change in weight; the data do not reflect dogs that may have initially lost weight and then regained it during the study period. This presents an opportunity for further study. Another limitation is the impact of COVID-19 restrictions during the study period and how these may have impacted canine weight.

### 4.2. Veterinary Engagement

Following AAHA weight management guidelines as summarized in Table 2, VE0-VE2, as measured in this study, do not meet the AAHA standard of care but a score of VE3 is considered consistent with the AAHA recommendations. The results of the review of 2437 EMRs show that VE0–VE2 occurred 83.3% of the time, with the highest level VE3 occurring 16.7% of the time. As shown in Table 1, the AAHA guidelines are extensive, and consistently engaging at VE3 may not be feasible during a typical wellness or sickness exam.

Overweight and obesity are a growing disease affecting millions of dogs, and recent trends show an increase in the prevalence of obesity (21%) among growing dogs, which further supports the importance of discussing weight management before dogs become obese [61,62,63,64]. Regardless of age, breed, and sex, the mean weight change was +2.0% based on the weight at initial diagnosis compared to the final weight entry. All levels of VE correlated with weight gain.

A starting BCS of 6 was the most frequent starting measure (46.2%; *n* = 220), followed by 7 (41.0%; *n* = 195). VE did increase linearly with increased BCS and was highest at a BCS of 7. While this does suggest that weight gain may result in an increased likelihood of VE, it also suggests that veterinarians are missing an opportunity to engage with clients about weight earlier at a BCS of 5 or 6. As shown in Figure 4, VE at all levels was quite low for BCS 6. These findings support the opinion of Kipperman and German that veterinarians may not be speaking up about overweight and obesity each time they are recognized [47]. But in reality, communicating the importance of weight management can be challenging. Veterinarians struggle to effectively communicate with and educate clients on the importance of weight management. They often fail to convince them to adhere to a weight management program. The subject of weight can be a sensitive topic and may be uncomfortable for veterinarians who fear this information may offend or anger a client [65,66].

A 2022 survey documented communication barriers for small animal veterinarians discussing nutrition with clients. These barriers included client resistance to a diet change, veterinarian time constraints during both well and sick pet appointments, nutrition misinformation online, and difficulty keeping up with the rapidly changing and expanding market for pet foods [64]. It may also be that veterinarians approach the problem as a lifestyle choice rather than discussing overweight or obesity as a disease. The findings of this study are also consistent with a 2018 analysis of weight management communication of 284 veterinarians in Ontario, Canada, which found that 60% of veterinarians discussed nutrition and only 12% recommended long-term dietary changes for weight management [65].

In many instances, even effective communication and high levels of VE can fall on deaf ears if the client is disinterested in the subject. Owner behavior, lifestyle, and human–animal interaction are powerful and make the management of this disease even more difficult. The human–canine relationship is very strong and can supersede any effort to achieve weight management. Many owners feel so strongly about the importance of providing food and treats in their relationship with their dog; even if they acknowledge their dog needs to lose weight, they are unable to change their own behavior. This exemplifies the complexity of canine overweight and obesity. Most dogs will naturally seek food and treats, and their enthusiastic response to food provides positive reinforcement for both owner and pet, and the cycle continues. The dog learns to beg for food/treats and the owner then becomes accustomed to providing what he/she thinks will make the dog happy, even if it negatively affects their health and welfare.

In this study, several different veterinarians examined the same patient often with different assessments of BCS and ideal weight calculations. A review of EMRs that showed BCS scoring, which is a subjective measure, was variable and at times contradictory in diagnosing overweight and obesity for the same patient. This variation in assessment may result in client confusion when trying to follow veterinary recommendations for weight management. Additionally, BCS and IBW were not calculated and communication with the client about weight was not consistently documented at each exam. For example, there were several patients with some EMRs without a recorded weight and BCS. This may contribute to a failure to adequately diagnose, treat, and prevent this disease. These findings align with other studies, suggesting veterinary professionals are underreporting overweight and obesity status and could be missing key welfare opportunities [12,66]. These results also support the need for exploring more precise tools to score body condition [67]. Additionally, establishing predictors of disease based on BCS may increase VE and client compliance.

This study showed that when dogs were assessed at a BCS of 7, VE3 was highest. At a BCS of 8, overall VE increased, but VE3 decreased, with similar findings at a BCS of 9. This suggests that obese dogs may not receive the level of veterinary engagement required. One possible explanation may be that a history of client non-compliance can lead to veterinary weight loss discussion fatigue. Other possible factors as described by Kipperman and German include a desensitization to obesity, prioritizing the client relationship over the patient’s health, or assuming clients will fail to comply with a weight loss plan [47]. If a dog has been diagnosed as overweight or obese for years with no measurable weight loss, veterinarians may begin to assume clients do not believe weight loss is an important disease and further discussions will be ineffectual.

In some instances, pet owners simply may not be ready to commit to a weight management program and need more time to accept the diagnosis. This lack of immediate buy-in from clients may lead veterinarians to conclude that owners either are not interested in weight management, do not agree with the diagnosis, or believe this is a disease with serious health implications. As a result, veterinarians may place less emphasis on discussing the importance of weight loss with each subsequent veterinary exam. By comparison, when a dog is diagnosed with renal or cardiac disease, veterinarians make very explicit recommendations for disease management. This does not appear to be the case with overweight and obesity, which supports the need to classify this as an important chronic disease [11].

Of significant importance and not measured in this study is the degree of client compliance at each VE. Such information is difficult to obtain in a retrospective study of EMRs but given that lack of client compliance is considered one of the most important human factors in canine overweight and obesity, gathering such data would prove valuable to further understanding this disease.

VE is critically important in fostering favorable welfare states and the diagnosis, management, and prevention of overweight and obesity. The results of this study show veterinarians inconsistently diagnose and engage with clients to address the disease. This suggests that while there are several useful resources and tools available, their use has not become the standard of care and many veterinarians do not have the time required to follow these recommendations as part of a regular well or sick patient examination. The continued increase in disease prevalence supports the need for further research to develop more effective strategies to improve communication to achieve successful weight management.

### 4.3. Prescription Weight Loss Diets

Studies confirm RX diets increase weight loss, yet as this study confirms, their use is low [7,29,68]. In this study, RX diet use was 8.4% (*n* = 40), and when used, 57.5% (*n* = 23) of dogs lost weight and 15.0% (*n* = 6) of these dogs achieved a BCS of 5. This low use of weight loss diets corroborates with other studies, suggesting that RX diets are underutilized [56]. This may be due to the owner’s perception that RX diets are more expensive or the hospital is just trying to sell them a particular expensive food [69]. Owners may also be concerned that diet food will not taste good or that their dog will not get enough to eat and beg them for more food. But Hours et al. conducted a palatability study and found RX diets to be highly palatable with no significant difference in consumption between different RX diets [70]. And German et al. concluded that RX diets are cost-neutral when compared to average food costs [71]. Another possible reason for low RX diet use is veterinary reluctance to recommend these diets until dogs are at a higher BCS. Client reluctance to accept the use of RX diets until their pet is obese is another factor, with clients preferring to “cut back on table scraps” or “reduce treats” rather than change their dog’s diet [54].

Several studies suggest clients want nutrition advice from their veterinarian but when surveyed, they also seek advice from other sources like breeders, online, or the pet store [10,43,52,72]. The patient records reviewed in this study included little to no diet history or nutritional recommendations. This information void appears to be filled by other less reliable and often confusing sources. Pet owners place significant importance on the quantity and perceived quality of food offered and are often confused by the tremendous variety of foods available, only made worse by the lack of label regulations. Clients may be drawn to cleverly marketed over-the-counter diets labeled as “less active”, “weight control”, “optimal weight”, or “healthy weight”, with no evaluation of the caloric density. These diets may still be too high in fat and calories but carry these labels because they are lower compared to the company’s other products [10]. The lack of strict labeling practices is misleading and owners who believe they are working on weight management may instead continue to overfeed their dogs.

This study’s low documented use of RX diets may be an underreporting of their use since the EMRs reviewed do not consistently document current diets being fed and not all RX diet purchases show up as a prescription in the EMR. Another possible explanation is owners often fail to comply with weight loss diet recommendations and such recommendations may not always be documented in the EMR. But despite a low rate of RX diet use (8.4%), prescribing an RX diet did result in weight loss. These results are consistent with other studies which show that RX diets result in more weight loss [3,53]. This is an area requiring further investigation and is an opportunity for weight loss success and ongoing weight management.

### 4.4. Comorbidities

Overweight and obesity contribute to multiple serious chronic disease processes by increasing the risk and severity of many diseases, reducing quality of life, and shortening lifespan by almost two years [18,21,31,32,33]. The high incidence (49.4%) of weight-related comorbidities in this study is consistent with other studies [17,23,25,73,74]. Almost half the patients in this study were diagnosed with one or more diseases falling into six comorbidity categories with orthopedic, dermatologic, and cardiovascular diseases as the top three, followed by endocrine, neoplasia, and renal/urinary.

Orthopedic disease is a broad category of painful conditions and was the highest reported weight-related disease (29.6%). OA and DJD are the more commonly diagnosed orthopedic diseases and result in chronic pain and reduced mobility, which contribute to muscle atrophy and weakness. The incidence of OA is increasing and, according to Banfield, has increased by 66% in the past 10 years, and 52% of dogs diagnosed with OA are also overweight or obese [75]. A 2021 report indicated that more than 150,000 dogs are diagnosed with OA every year [17].

Dermatologic disease includes a variety of conditions that may be more severe in overweight and obese dogs. Obesity causes a state of chronic, low-grade inflammation directly related to excess adipose tissue [76]. Most dermatologic diseases involve chronic inflammation. Some conditions such as skin fold pyoderma can be worse in overweight and obese dogs; 13.2% of dogs in the study group had dermatologic disease.

While overweight and obesity are not typically considered a risk factor for developing heart failure in dogs, they can compromise cardiopulmonary function and worsen the clinical signs of heart disease [77]. According to a study by Pereira-Neto et al., when compared to dogs with an ideal BCS, obese dogs had a lower arterial partial pressure of oxygen, a decreased tidal volume and inspiratory and expiratory time, and an increased respiratory rate [60]. Overweight and obese dogs have higher anesthetic risks, making preventive care such as routine dental cleaning more dangerous. Dogs with conditions such as a collapsing or hypoplastic trachea and brachycephalic breeds tend to have more severe respiratory issues and cough more when they are overweight or obese. In total, 11.6% of dogs in the study group had cardiovascular or respiratory disease.

Other comorbidities occurred less frequently and included endocrine diseases at 4.4%, neoplasia at 3.4%, and renal and urinary diseases at 2.9%. In humans, obesity is linked to cancer, with many types of cancer developing secondary to chronic inflammatory conditions [76]. Overweight and obese dogs suffer from an array of comorbidities. This disease should be treated with as much urgency as other serious life-threatening diseases [10,21,45]. Clients react differently when their pet is diagnosed with what is considered a well-known or real disease as opposed to obesity. For example, when their pet is diagnosed with heart disease, renal disease, or cancer, pet owners take it very seriously and will generally follow veterinary recommendations. As the review of the results in this study shows, this does not appear to be the case when an owner is informed their dog is overweight or obese. As suggested by the Global Pet Obesity Initiative, it may be due to the fact that obesity has not been considered a disease and is thus not taken as seriously as a diagnosis of heart disease, renal disease, or cancer [4]. Possible explanations for this stem from our own challenges with weight management and body image. The divergence in public opinion about weight, with the impact of social media opinion ranging from pressures to obtain the perfect body to accepting being overweight or obese as expressing body positivity, may influence how we look at our pets. Another consideration is the public has become accustomed to seeing overweight dogs to the extent they believe overweight dogs look healthy. As described by Corbee in a study of show dogs, a BCS of 6 to 7 was considered the breed standard almost 20% of the time, and it was concluded that there need to be ”…different interpretations of the [official breed] standards to prevent overweight conditions from being the standard of beauty” [78] (p. 909).

Overweight and obesity is a chronic, complex yet preventable disease negatively impacting canine welfare. VE is critically important in the diagnosis, management, and prevention of all diseases including overweight and obesity. RX diets have been shown to result in greater weight management success. Levels of VE and RX diet use were lower than expected in this study and present opportunities for improvement in canine weight management. This study’s results showed a high incidence of comorbidities associated with overweight and obesity increasing the risk and severity of multiple comorbidities, reducing both longevity and quality of life, and creating a tremendous financial and emotional burden on pet owners.

These results prove the need to explore other avenues of support and implement more efficient and practical tools to successfully manage this complex yet preventable health and welfare problem. To start, veterinarians should make the assessment of patient weight and BCS part of every examination starting as early as the first puppy visit and inform pet owners of the importance of maintaining a healthy weight throughout their dog’s life. Additionally, they should begin to discuss weight in the context of welfare with a focus on disease, stressing the deleterious effects of overweight and obesity on overall health, welfare, and longevity. Other areas of opportunity include an increased use of RX diets and making the recommendation sooner, such as at a BCS of 6, to help prevent obesity. Veterinary hospitals can streamline protocols based on the AAHA recommendations and consider scheduling designated weight management exams. Finally, the use of technology may be a valuable tool to help educate clients on the importance of proper nutrition and increase compliance with veterinary weight management programs.

## 5. Conclusions

Given the large number of dogs experiencing poor welfare due to overweight or obesity and the negative effects on health and quality of life, this may be the most important welfare problem for canine companion animals. The results of this study reveal that early and consistent VE regarding canine weight management is lacking. Although there is not a strong relationship between VE levels and weight loss, this is an area with a significant opportunity for improvement. RX diets are underutilized as a safe and effective weight loss and management tool and are another area where improvement is needed. Veterinarians are ethically and professionally obligated to protect animal health and welfare and prevent disease, yet veterinarians do not consistently document a diagnosis of overweight or obesity and have limited success in helping patients achieve their IBW. These results suggest veterinarians struggle to effectively communicate that overweight and obesity are serious diseases and educate clients on the importance of weight management. As a result, the overweight and obesity epidemic continues to compromise the health and welfare of our canine companions.

## Figures and Tables

**Figure 1 animals-14-00740-f001:**
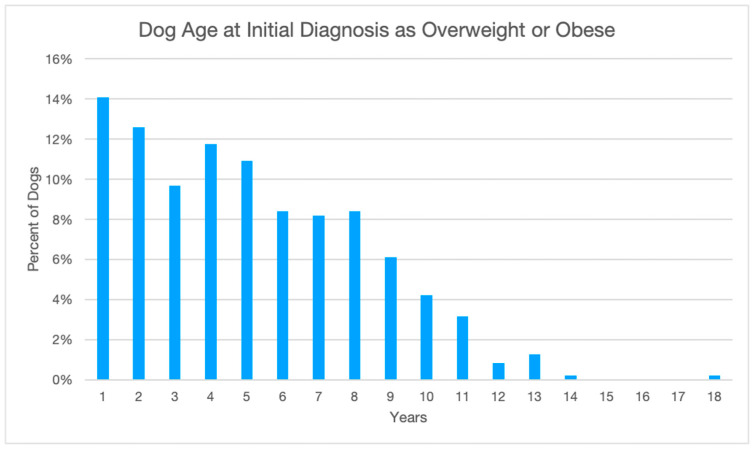
Average age of patients when initially diagnosed as overweight or obese. *n* = 476.

**Figure 2 animals-14-00740-f002:**
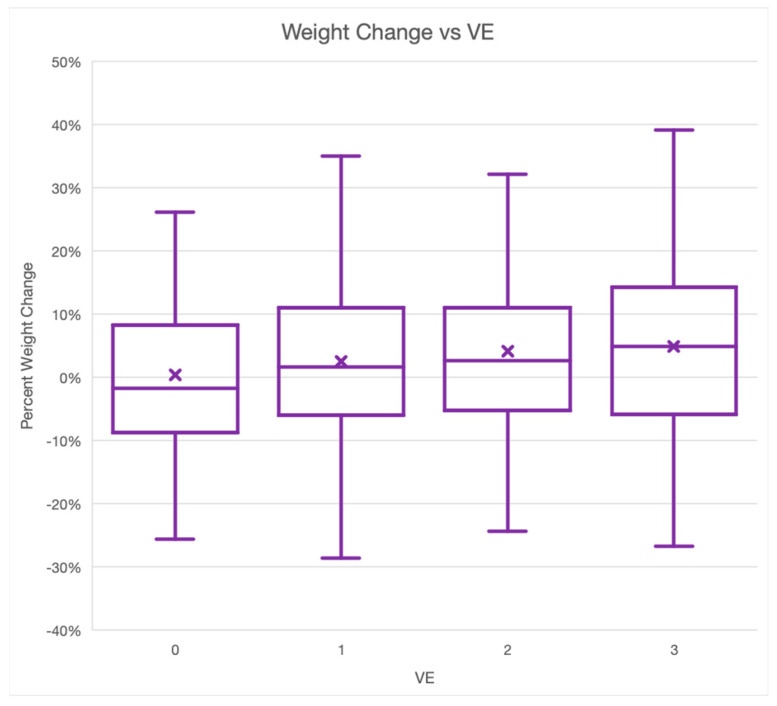
Box and whisker plot of the relationship between VE and percentage weight change. *n* = 476.

**Figure 3 animals-14-00740-f003:**
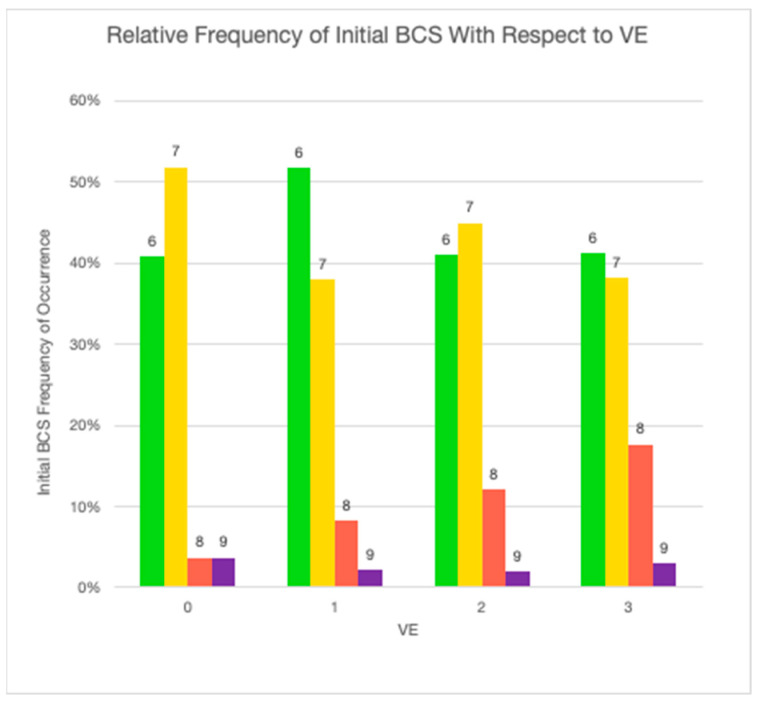
Relative frequency of initial BCS with respect to VE (green: BCS = 6; yellow = 7; red = 8; purple = 9).

**Figure 4 animals-14-00740-f004:**
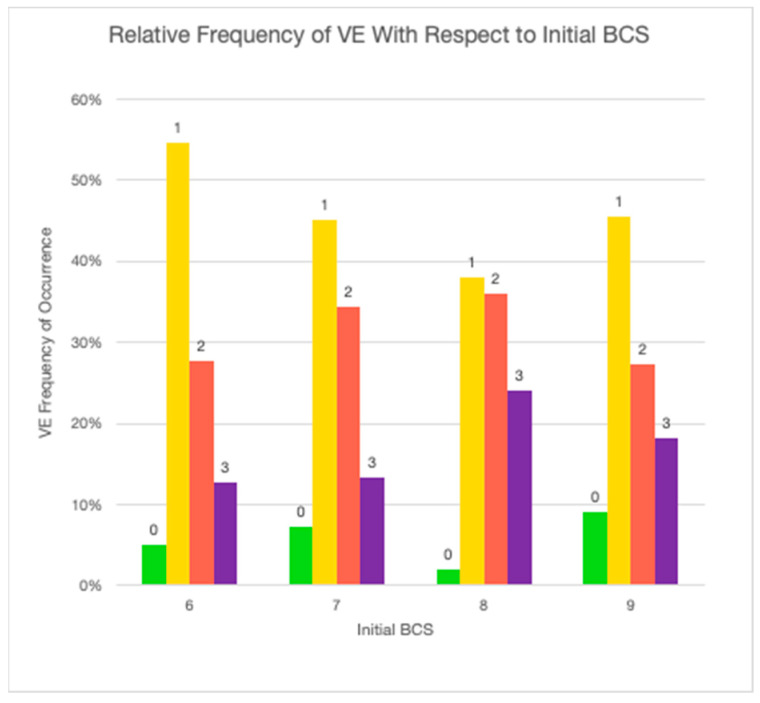
Relative frequency of VE with respect to initial BCS (green: BCS = 6; yellow = 7; red = 8; purple = 9).

**Figure 5 animals-14-00740-f005:**
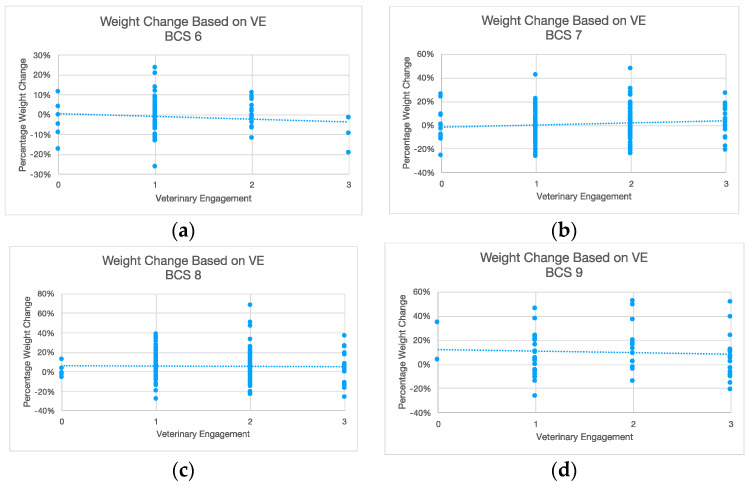
Weight change based on VE for each BCS. (**a**) BCS = 6; (**b**) BCS = 7; (**c**) BCS = 8; (**d**) BCS = 9.

**Figure 6 animals-14-00740-f006:**
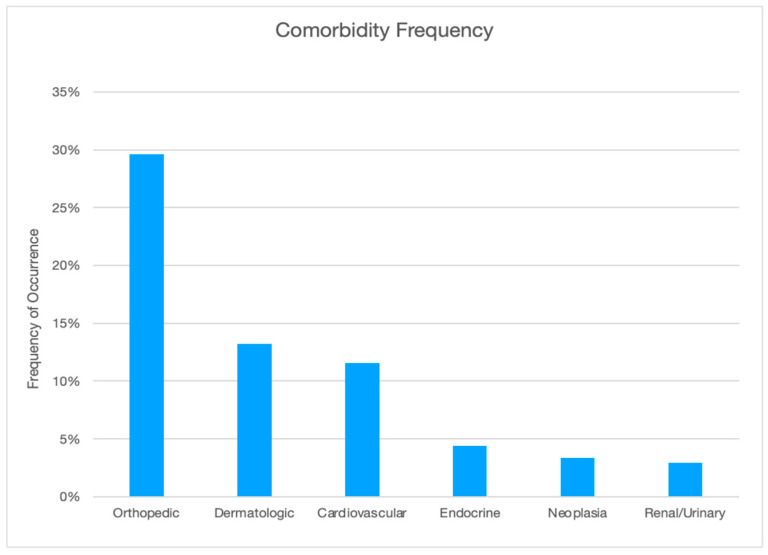
Frequency of comorbidity by category.

**Figure 7 animals-14-00740-f007:**
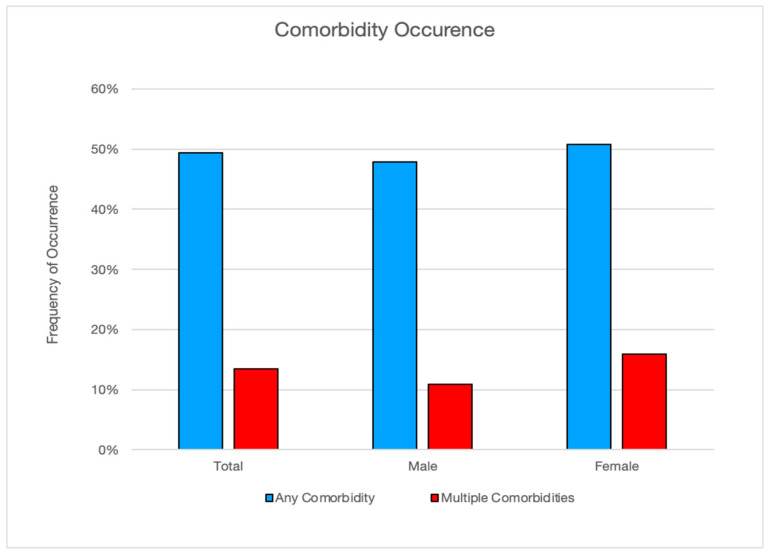
Frequency of comorbidity by sex.

**Table 1 animals-14-00740-t001:** Summary of AAHA 2014 weight loss program design.

Assess current weight and BCS (and ideally MCS)
Calculate ideal body weight
Obtain a detailed diet history
Determine current daily caloric intake
Calculate the daily caloric requirements for weight loss using the following formulas:
○ RER in kcal/day = 70 × (ideal BW [kg])^0.75^ (for all dogs)
○ RER in kcal/day = 30 × (ideal BW [kg]) + 70 (for dogs between 2–25 kgs)
Calculate the safe rate of weight loss (1–2% body weight/week)
Make precise diet and treat recommendations
Prepare detailed feeding management and exercise plans
Schedule follow-up appointments
Adjust the plan as needed
Create a plan for ongoing maintenance once goals are achieved

**Table 2 animals-14-00740-t002:** VE score and associated description of documented engagement with clients.

Level of Engagement Score	Description of Engagement
0	Diagnosis only. No documented discussion with client in MR.
1	Basic recommendation of “…eat less and exercise more”.
2	determination of percent and amount overweight. ±ideal body weight (IBW) calculated. recommendation of percentage caloric reduction. increase activity.
3	Recommendation of 1 and 2 above plus the following: calculation of current caloric intake, kcal/day to feed. ±actual measurement of food to feed. ±recommend weight rechecks. ±RX diet recommendation. ±written instructions.

**Table 3 animals-14-00740-t003:** Comorbidity classification and diagnoses.

Comorbidity Classification	Diagnoses
Orthopedic	Osteoarthritis, lameness, intervertebral disc disease, cruciate ligament tear, patella luxation, neck/back pain, dysplasia, reluctant to walk, mobility issues
Dermatologic	Atopy, chronic dermatitis, chronic otitis, pruritus, recessed vulva, chronic anal gland infections
Endocrine	Hyperadrenocorticism, hypoadrenocorticism, hypothyroidism, diabetes mellitus
Renal/Urinary	Recurring urinary tract infections, chronic kidney disease, proteinuria, urolithiasis, urinary incontinence
Cardiovascular/Respiratory	Brachycephalic obstructive airway syndrome, collapsing trachea, heart murmur, respiratory distress, chronic cough
Neoplastic	Lymphosarcoma, hemangiosarcoma, anal gland adenocarcinoma, carcinoma, chondrosarcoma, mammary carcinoma

**Table 4 animals-14-00740-t004:** Baseline information of patients included in this study.

Variable	Count	Percent
Number	476	
Sex		
Female neutered	236	49.6%
Female intact	10	2.1%
Male neutered	221	46.4%
Male intact	9	1.9%
**Age (months)**		
Average beginning	62	
Average ending	98	
**Top Breeds**		
Retriever, Labrador Mix	63	13.2%
Chihuahua Mix	48	10.1%
Shepherd Dog, German/Mix	26	5.5%
Terrier Mix	26	5.5%
Dachshund Mix	19	4.0%
Retriever, Golden	19	4.0%
Maltese Mix	17	3.6%
Mixed Breed	17	3.6%
Pitbull Mix	17	3.6%
Beagle/Beagle Mix	15	3.2%
Poodle Mix	12	2.5%
Shih Tzu	12	2.5%
All other	185	38.9%
**Starting BCS**		
6	220	46.2%
7	195	41.0%
8	50	10.5%
9	11	2.3%
**Ending BCS**		
4	1	0.2%
5	49	10.3%
6	138	29.0%
7	174	36.6%
8	86	18.1%
9	28	5.9%

**Table 5 animals-14-00740-t005:** Level of veterinary engagement (VE) and change in patient weight.

VE Level	Overall	Lost Weight	Gained Weight	Neither	Overall Weight Change
**VE 0**	5.7%	55.6%	40.7%	3.7%	0.3%
Sample size	27	15	11	1	
% Weight change		−0.18% to −25.68%	0.30% to 34.24%		
**VE 1**	48.7%	44.4%	53.9%	1.7%	2.5%
Sample size	232	103	125	4	
% Weight change		−0.1% to −28.7%	0.1% to 45.5%		
**VE 2**	31.3%	43.6%	55.7%	0.7%	4.1%
Sample size	149	65	83	1	
% Weight change		−0.4% to −24.3%	0.2% to 67.2%		
**VE 3**	14.3%	35.3%	60.3%	4.4%	4.9%
Sample size	68	24	41	3	
% Weight change		−0.7% to −26.7%	0.8% to 51.4%		

## Data Availability

Data supporting reported results can be found at https://osf.io/s27et (accessed on 15 February 2024).

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
