# Peer review of "How Successful Are Veterinary Weight Management Plans for Canine Patients Experiencing Poor Welfare Due to Being Overweight and Obese?"

_animals, 2024, doi:10.3390/ani14050740_

Round 1
Reviewer 1 Report
Comments and Suggestions for Authors
Overweight and obesity management in companion animals is a current topic due to the health complications that it can cause and the costs that it can represent for the owner due to medical care. For this reason, I consider that this study is innovative and addresses a topic that concerns veterinarians. However, a weakness of the manuscript is the objective that, although presented, should be more concise so that the reader can understand the broad framework.
Particular comments
Lines 7 – 11. I recommend moving these lines to the Abstract (in line 20) since the simple summary has a word limit.
Lines 26 – 28. I recommend using the same aim of the study that was mentioned in lines 11-13.
Line 29. The author could include information about the number of evaluated breeds, their average age and body weight, etc.
Lines 30 – 31. Could the authors mention why overall weight loss was low? For example, if the average weight loss was some percentage.
Line 33. Please, consider complementing the keyword “welfare” with “animal welfare”, and add “nutrition”.
Lines 36-44. Please, consider using more references apart from German et al., 2016.
Line 67. Here, the authors could briefly mention that worldwide issues such as COVID-19 quarantine might also affect the presentation of obesity in animals.
Lines 75 – 83. I recommend adding a hypothesis for the present study.
Section 2. I appreciate the effort of the author by adding this detailed section regarding the definition, causes, and implications of overweight in companion animals. However, I highly recommend summarizing this information and combining it into the Introduction or the Discussion section.
Lines 401 – 406: Since the aim of the study was already mentioned at the end of the Introduction, I do not consider it necessary to include these lines.
Lines 426 – 441. I recommend re-structuring this paragraph as follows:
· Lines 431 – 436. Both sentences are similar to what was already described in lines 414-415 and 420-425. My recommendation is to just state that 17 patients with only one record and 7 with chronic diseases were excluded. It is not needed to explain again why these patients were not included.
· After mentioning that 476 patients were included, mention that from these patients, a total of 2.437 EMRs were reviewed, in the period from 01 January to 24 August 2022. These medical records represented an average of 5.12 records per patient, where some patients had a minimum of two and a maximum of nine weight and BCS measurements.
· Then after mentioning this, describe that each medical record was reviewed for weight, BCS, veterinarian engagement level, prescription weight loss diet, and comorbidities.
Section 3.5. Comorbidity assessment. I suggest adding if this assessment was considered only in the first record of the overweight patient, or if it could be a comorbidity that they had before their first record or that was diagnosed during the consecutive records of the same patient.
Figure 3 is not mentioned in the text, and I also do not think that this graph provides additional information that hasn’t been already included in Table 5.
Figures 4 and 5. There is no mention of the meaning of the bars with the different colors (I understand that it corresponds to BCS of 6, 7, 8, and 9 but it is necessary to add it directly to the image or in the footnote).
Figure 6. The text that appears in line 615 should be written just after the Figure title.
Figure 7 is not mentioned in the text, and, from my perspective, it does not show any different information than Figure 6. I would recommend leaving one or another.
Figure 8 is not mentioned in the text.
Lines 707 – 717. This information was already mentioned in the Material and Methods section. Consider deleting this paragraph.
Lines 721 – 724. Are there studies reporting why these breeds are susceptible or predisposed to obesity? Could it be something genetic? If there are studies about it, I recommend briefly discussing this.
Lines 378 – 742. I also think that another factor that may be considered as a limitation or that could have contributed to dogs not losing weight in the period that the author states (2017-2022) could be related to the COVID-2019 pandemic. During lockdown, owners and companion animals were not able to exercise or walk as much as before the pandemic. The authors could see Vučinić et al., 2022 https://doi.org/10.1016/j.jveb.2021.10.009 and Owczarczak-Garstecka et al. 2021 https://doi.org/10.3390/ijerph18126315 studies.
Lines 893 – 901. Another factor that was mentioned in the introduction is the owner’s lifestyle. This was discussed in Mota-Rojas et al. (2021)’s review https://doi.org/10.3390/ani11113263 and also has been reported by Muñoz-Prieto et al., 2018 https://doi.org/10.1038/s41598-018-31532-0 and Linder et al., 2021 https://doi.org/10.3389/fvets.2021.654617
Line 936. Correct “pf” to “of”.
Reviewer 2 Report
Comments and Suggestions for Authors
This paper addresses the worthwhile and widespread issue of overweight and obese dogs. In general, there was decent scientific rigor evident in the study, especially since statistical analysis was undertaken, which is not always common in papers of this nature. However, overall, this paper would benefit from significant revisions. Specific recommended edits are described below. In general there was a lot of redundancy and the unnecessary representation of information throughout. The manuscript is quite long, and though it's understandable that a study of this scope would result in a manuscript on the longer side, it can be significantly tightened up to ensure information is presented in a concise and understandable manner. Additionally, there are areas of the paper that should be expanded upon, such as issues raised (or not raised) in the Discussion.
Simple summary
Lines 7-8: the first sentence doesn’t flow particularly smoothly so rewording should be considered; at a minimum “is considered” should be changed to “are considered”. This should be changed throughout the paper in order to be grammatically correct. (Overweight and obesity are two nouns, and thus “are”, not “is” should be used.
Line 16: there should be a comma between “obesity” and “or”.
The Abstract and Simple Summary should be revised to include brief mention of the type of analysis done, namely statistical analysis. Both should better reflect the key findings of the study.
Introduction
Lines 45-47: this sentence doesn’t make a lot of sense and can be streamlined, so it could be reworded to something along the lines of, “…where a score of 1 is emaciated and 9 is morbidly obese, with a score of 5 being ideal an ideal weight.”
Line 51: start a new sentence with “Overweight and obesity…”.
Lines 86-93: all of the information in this paragraph has already been presented in the opening paragraph of the Introduction, and it does not need to be repeated. Since this subsection is for definitions, it would make sense to remove it from the introduction and leave it here.
Line 101: when writing body scores, they can just be written as the single number, not x/9, since it has already been established that it’s a 1-9 scale. This should be changed throughout the manuscript.
Line 106: apostrophe needed – should be “observers’ visual…”.
Line 118: it seems unclear what “BCS -5/10” refers to and how they figures into the equation based on the example given below.
Lines 134-139: this information has previously been presented and does not need to be repeated here.
Line 169: it would seem the fact that owners also control the quality/type of food dogs eat also plays a role. Even if this isn’t stated in the cited source, note could me made of that possibility in a separate sentence.
Lines 177-179: this sentence is a bit unclear and should be reworded to something along the lines of “…owner feeding habits and strategies, such as using food as a reward and to show affection, and lack of exercise…”
Lines 190-192: this sentence or study finding seems confusing. As is, it would seem to mean that dog owners rated their own dogs as overweight or obese on the scale, but still didn’t feel their own dog was overweight – even after rating them as such. If that interpretation is correct, then aside from possibly viewing such dogs as “normal”, it would also suggest that the rating scale is lacking validity, or possibly there’s a disconnect or dissonance in owners’ ability to accurately assess their dogs’ weight.
Lines 208-210: this sentence has been stated multiple times previously, and does not needed to be repeated here, or at least in its current wording.
Line 264: should be “affects”, not “effects”.
Lines 268-269: not sure how it can increase the risk of anesthesia, so perhaps this sentence needs to be corrected.
Lines 292-294: numerous studies have found that behavior reasons are the number one reason group of dog relinquishments, so it might be worthwhile to cite some of those.
Line 298: the beginning of this sentence doesn’t seem to make any sense and should be revised, “…such as may result from…”.
Materials and Methods
Lines 444-446: the BCS scale has already been explained and does not need to be reexplained here.
Line 447: should be “were entered”.
Line 500: Oscar should be in all caps.
Lines 517-519: these are results (from the regression analysis), and should go in the Results section.
Results
Table 5: title of table has a formatting issue and part of it ended up below the table itself.
Figures 1 & 3: sample size should be noted in figure titles.
It would be interesting and potentially worthwhile to investigate whether there is relationship between breed and weight loss success.
Discussion
Lines 686-695: this information appears previously in the manuscript and does not need to be repeated again here.
Lines 705-717: just as with previous comment, this information does not need to be repeated here.
Line 889: this has been noted previously in these comments, but the top BCS score of 9 does not need to be given each time a BCS is mentioned (e.g. 7/9). It has been well established early in the manuscript that it’s a 0-9 scale and the repeated noting of it can cause a decrease in comprehensibility by the reader.
There should be mention of how the nature of the sample may have affected this study’s findings, especially since there was homogeneity in the vet hospitals included in the study. There might be a uniqueness to the type of clients who visit Banfield hospitals compared to elsewhere, such as small, community vet practices.
It would be worthwhile in this section to discuss the potential impact of various facets of the dog-owner relationship as a cause for overweight and obese dogs. Such aspects were briefly noted earlier in the manuscript, but they should be given greater attention in this section. For example, the likelihood of owners using food/treats as a way to bond with their dogs or show them love/attention. Owners may also feel that by giving their dogs either an excess quantity of food/treats or allowing them to overindulge in certain types of food that is not nutritionally beneficial, they are “spoiling” their dog. And this suggests that by “spoiling” their dog they are doing something that is in the best interest of the dog, when in fact, as described in this paper, such behaviors can lead to the development of weight-related health issues.
Conclusion
Lines 914-917: based on the results of this study, it seems that although VE levels are lacking in regard to weight management, there’s not a strong relationship between VE levels and weight loss. Therefore, low VE levels are rather a moot point as they affect weight loss.
Lines 928-941: the recommendations section should be moved into the Discussion, and a very brief mention of them should be added to the Conclusion.
Lines 931-935: it would also be a worthwhile suggestion for vets to discuss the health risks associated with overweight and obese dogs for all dogs, not just those who are already overweight, as at that point, meaningful intervention might be more challenging. Instead, a discussion around the importance of maintaining a healthy weight should be prospective.
Comments on the Quality of English Language
There are several grammatical and sentence structure issues throughout the manuscript, including long/run on sentences. The addition of commas in many sentences would also make the information presented more comprehensible by the reader. There were too many of such issues to note each one individually, but a thorough review and revision of the sentence structure in particular is recommended.
Reviewer 3 Report
Comments and Suggestions for Authors
How successful are veterinary weight management plans for 2 canine patients experiencing poor welfare due to being over-3 weight and obese?
Manuscript ID: animals-2852333
Summary
This study attempts to determine the extent and effectiveness of current veterinary weight management guidance by assessing the level of veterinary engagement and its impact on patient weight loss.
General concept comments
Article
It is an interesting idea to do a study. Perhaps with some adjustments one could make better use of all the available data.
Review
While it is an interesting idea to explore the relationship between the level of veterinarian engagement in the problem of overweight and obesity in dogs, the methodology lacks objectivity by overlooking such important factors as the level of owner engagement. The way in which veterinarians' engagement is assessed is too subjective and inadequate, so the results are not entirely accurate. In addition, the manuscript is generally too long and repeats many ideas throughout the text, and many of the references are more than five years old.
Specific comments
16. Please check for extra spaces between words.
26. Same as line 16.
59. Check the extra spaces throughout the text.
86-102. You repeat the first paragraph of the Introduction (lines 36-47).
134-139: Again, you are repeating information already covered in the Introduction (lines 48-57).
84-399: The Introduction should cover both the background and the purpose of the study. It would be clearer if the two sections were combined. In addition, it is necessary to summarize all the Background information as it is too extensive, and many ideas are repeated in both sections.
401-406: This paragraph should be at the end of the introduction.
512-514: How was the effect of veterinary engagement separated from owner influence? The veterinarian may have made recommendations for weight change, but the owners may not have followed those recommendations. Therefore, it may be the case that there was not a relevant weight change because the owners did not follow through.
539-638: This is because prescribing a diet for weight loss does not necessarily mean that the owner will follow the diet for their pet. It is not an adequate way to measure the effectiveness of a diet or veterinary engagement.
686-695: Again, you are repeating information already covered in the Introduction (lines 48-57).
705-716: This is part of Materials and methods. It is not necessary to repeat it.
685-925: While there is evidence that veterinarians care about overweight and obese dogs, the way this study measures engagement is flawed and subjective. Perhaps the methodology should focus on the gap in communication between veterinarians and owners. The fact that dogs are not losing weight is not necessarily due to a lack of veterinarian engagement. How owners understand the instructions has a big impact.
980. Check the format of the references. There needs to be consistency. Also, some are more than five years old.
Comments on the Quality of English LanguagePlease check for extra spaces between words throughout the text.
Round 2
Reviewer 1 Report
Comments and Suggestions for Authors
The author has made substantial changes to the previous version. Furthermore, they have addressed most of the requested adjustments. I believe the document now reads better and is more seamlessly integrated. I respectfully give my authorization for its publication.
Reviewer 2 Report
Comments and Suggestions for Authors
Thank you for making the suggested revisions. I know it can be an overwhelming process to write and submit your first paper. I think the changes you've made have improved the quality of the paper. However, there are still some minor typos, grammatical errors, unnecessary capitalizations, and italicized words, specifically within the revisions. As such, my final recommendation would be to have one more read through it to ensure all of these tiny issues are fixed. For example, in the first line of the simple summary, there's a typo and it should be "is considered". Also, in the Discussion, it should be "dogs' lives", not "dogs' life".
Comments on the Quality of English LanguageSee above in the general suggestions.
Reviewer 3 Report
Comments and Suggestions for Authors
The text improved significantly after adding information about the subjectivity of the study and the difficulty in distinguishing between the influence of the veterinarian and the level of owner involvement.
Thank you for following the recommendations.
Success!
There are still some minor errors in the wording. A full revision is recommended.